# Association between obesity and urinary incontinence in older adults from multiple nationwide longitudinal cohorts

Xiyin Chen[1,2,6], Shaoxiang Jiang[3,6] & Yao Yao[4,5✉]

## Abstract

**Background** Obesity and urinary incontinence (UI) among older adults, particularly older men, are yet to be fully explored. Utilizing multiple nationwide prospective longitudinal cohorts representative of the US, UK, and European samples, we examined the association of body mass index (BMI) and waist circumference (WC) with UI among both older women and men.

**Methods** We derived the data from multiple longitudinal cohorts that surveyed UI. Participants were asked if they had experienced urine leakage within the past 12 months or within the past six months. The measure of obesity was based on BMI and WC. We employed a random-effect logistic model to associate BMI and WC with UI, adjusting for covariates including age, race, education, residence area, marital status, number of children, smoking, drinking, hypertension, diabetes, cancer, stroke, functional ability, and cognitive impairment. We visualized the associations by using restricted cubic spline curves.

**Results** A total of 200,717 participants with 718,822 observations are included in the baseline analysis. Compared to those without UI, both female and male participants with UI demonstrate a higher BMI and WC. Among females, the fully adjusted models show linear associations between BMI, WC, and UI ($P$s < 0.001). However, we observe U-shaped associations of BMI, WC with UI among males. The lowest likelihood of having UI is found among male participants with a BMI between 24 and 35 kg/m$^2$.

**Conclusions** Interventions aimed at preventing UI among older adults must take sex into account. Weight loss intervention could be an effective treatment among older females who are overweight and with obesity as well as older males with obesity rather than all older males.

## Plain language summary

It is not well known if being obese poses a risk of urinary incontinence (UI) in older adults, especially in older men. We aim to address this question by analyzing three nationwide long-term studies conducted in the UK, Europe and USA. We found there was a direct link between increasing body weight and the likelihood of experiencing UI in older females. Whereas, older males who are considered clinically obese were more prone to experiencing UI. This suggests that weight loss intervention can be effective for treating UI in older females carrying extra weight and older males who are considered clinically obese. Our study highlights that sex should be taken into consideration when developing interventions for UI treatment in older adults.

[1] Department of International Health, Johns Hopkins Bloomberg School of Public Health, Baltimore, MD, USA. [2] School of Public Health, LKS Faculty of Medicine, The University of Hong Kong, Hong Kong SAR, China. [3] Institute for Global Health and Development, Peking University, Beijing, China. [4] China Center for Health Development Studies, Peking University, Beijing, China. [5] Key Laboratory of Epidemiology of Major Diseases (Peking University), Ministry of Education, Beijing, China. [6] These authors contributed equally: Xiyin Chen, Shaoxiang Jiang. ✉email: yao.yao@bjmu.edu.cn

A rapidly aging population, coupled with age-related illnesses, is a pressing concern around the globe. Urinary incontinence (UI) is a common geriatric syndrome that has been associated with adverse health outcomes and poor quality of life. It is estimated that 17–55% of older females and 11–34% of older males had UI, which places a substantial financial and care burden on society and families[1]. The direct costs of UI in 1995 were estimated at $26.3 billion for those older than 65 years in the USA[2]. Furthermore, the annual cost-of-illness estimate for urgency urinary incontinence (UUI) in Canada, Germany, Italy, Spain, Sweden, and the United Kingdom was €7 billion in a 2005 multinational study[3]. UI is also associated with a reduced quality of life (QoL) and self-esteem for those who have a stressful, incapacitating condition and a social isolation element, with sufferers reporting embarrassment, distress, and depression[4]. The caregivers of older adults with UI are also more likely to feel a strain both physically and mentally than those without UI[4]. Thus, identifying and managing modifiable risk factors of UI is of paramount importance.

Although obesity is a potentially modifiable risk factor well documented to be associated with the development of UI, most epidemiological, biological, and clinical research focuses on older women rather than older men[5–17]. Meanwhile, despite some studies focused on both sexes[18–25], few longitudinal research targeting the older population investigated the sex heterogeneity in patterns of associations between UI and obesity[20,21]. Moreover, there are varied obesity indices, including body mass index (BMI) and waist circumference (WC). BMI estimates general obesity established by the World Health Organization (WHO), while WC better describes central obesity, although it cannot discriminate visceral fat from subcutaneous fat. However, the relationship between obesity indices, i.e., BMI and WC, and UI in older men and women has not been thoroughly examined in prior longitudinal studies. Furthermore, although the EpiLUTS study comprised research subjects from three countries[20], samples in most prior findings were based on samples from one country or region, which led to poor generalization of the study's findings.

Extant evidence has reported an "obesity paradox" in which BMI is negatively associated with mortality, which indicates that being normal weight and overweight could be more beneficial to health than being underweight[26]. This paradoxical association of obesity with mortality has been inconsistent due to a discrepancy between BMI and central obesity[27]. However, whether this paradox applies to older women and men in terms of UI was largely unknown. Therefore, a study with a representative sample of older men and women from a global perspective should be conducted to reexamine the associations between obesity indices, including BMI, WC, and UI.

In this work, by utilizing multiple nationwide prospective longitudinal cohorts representative of the US, UK, and European samples, we contribute to the literature by analyzing the association between BMI, WC, and UI among older women compared to men. Accordingly, our results depict linear associations among older females, whereas U-shaped associations among older males, regarding UI–BMI and UI–WC, which illustrates the sex difference in the impact of obesity on UI and provides an indication of evidence for future intervention.

## Methods

**Study design and participants**. Data were accessed from three international cohorts of aging: Health and Retirement Study (HRS)[28], English Longitudinal Study of Ageing (ELSA)[29], and Survey of Health, Ageing, and Retirement in Europe (SHARE)[30], which were used to provide the representative sample and comparable measures on BMI, WC, UI, and other covariates, covering 21 countries including both developed and developing ones on two continents. ELSA received ethical approval from the London Multicentre Research Ethics Committee (MREC/01/2/91). The University of Mannheim's internal review board (IRB) reviewed and approved SHARE for waves 1–4. From wave 4 onwards, the ethics reviews were done by the Ethics Council of the Max Planck Society. The detail of the ethics approvals is available from the SHARE project's website (http://www.share-project.org/fileadmin/pdf_documentation/SHARE_ethics_approvals.pdf). The HRS was approved by the University of Michigan Institutional Review Board (HUM00061128). We have applied and obtained permission to access and utilize the three longitudinal datasets described above[31]. Since this study is only based on de-identified data, no further ethical review for conducting this secondary analysis was required by Peking University[32]. Given the availability of UI measurements and similar time ranges, we used data from the following time period in this analysis: 2010–2018 for HRS, 2010–2018 for ELSA, and 2004–2010 for SHARE. Participants who were younger than 50 years old were excluded. Following the exclusion criteria, 121,450 SHARE participants with 360,800 observations, 19,791 ELSA participants with 98,158 observations, and 42,132 HRS participants with 207,805 observations were analyzed.

**Data collection**. Data on urinary incontinence were collected through questionnaires. In HRS and ELSA, UI was assessed by asking: "During the last 12 months, have you lost any amount of urine beyond your control?" In SHARE, the dependent variable was constructed based on the question: "For the past six months at least, have you been bothered by incontinence or involuntary loss of urine?" The responses ranged from 0 = "No", which meant the respondent hadn't been bothered by any urinary incontinence in the time frame corresponding to the question, to 1 = "Yes", which was considered as the respondent had been bothered by any urinary incontinence in the time frame corresponding to the question. The measures of obesity indices were based on BMI and WC. The BMI ($kg/m^2$) was derived by dividing weight by the square of height. The measurement of WC was reported in centimeters. The obesity indices (BMI and WC) were then equally divided into six quantiles when analyzing the association of obesity with UI prevalence. Women and men, as well as males and females, are self-identified binary gender categories.

In addition, covariates for sociodemographic status (age, race, residence area, marital status, number of children), lifestyles (smoking, drinking), the history of the disease (hypertension, diabetes, cancer, stroke) as well as educational attainments, functional ability, and cognitive impairment were added. The race was divided into white or not. Residence area was attributed to whether living in rural or urban areas. Marital status was imputed to married/partnered or not. We used the harmonized education attainments classification, namely lower than the secondary/upper secondary and vocational training/tertiary. Physical active participants were defined as those without difficulty using the toilet. Cognitive impairment was defined as a score of 8 or below on a 35-point scale for HRS[33,34], while a score between 0 and 11 on a 27-point scale for the ELSA[35]. In SHARE, a summary cognitive function score of averaged z-scores of the verbal fluency, immediate and delayed recall, orientation, and numeracy for individuals was constructed using the mean and standard deviation of the first wave, where respondents were labeled as having cognitive impairment if their score fell within the lowest decile[36].

**Statistical analysis**. Separate descriptive characteristics of the observations in the HRS, ELSA, and SHARE were provided, in

which the mean (SD) or the median (IQR) was used for continuous variables, and number (percentage) was used for categorical variables. The observations of BMI, WC, and other covariates were then displayed by whether with UI in both females and males for HRS, ELSA, and SHARE, respectively. Additionally, the statistically significant difference between those with UI and without UI in BMI, WC, and other covariates was reported to identify the driving factors of UI by sex among older adults in the HRS, ELSA, and SHARE.

All statistical analyses were conducted by females and males separately. Chi-squared tests were used to evaluate differences among those with UI and without UI for categorical variables, whereas $t$-tests were used for continuous variables. We employed random-effect logistic models to control the individual-specific characteristics that might change in different waves and to construct odds ratios (ORs) and 95% confidence intervals (CIs) to analyze the relationships between obesity indices and the prevalence of UI. The dependent variable is the prevalence of UI. The sociodemographic status, lifestyle factors, history of diseases, educational attainments, functional ability, and cognitive impairment were incorporated as the co-variates. Three models were conducted in the analyses by sex. Model 1 was a crude model. In Model 2, we controlled for age, race (except for the SHARE), educational attainments, residence area (except for the ELSA), marital status, and the number of children. Model 3 was further adjusted for current smoking, ever-smoked, alcohol consumption, physical activity, hypertension, diabetes, stroke, cancer, and cognitive impairments, which was regarded as the fully adjusted model. Statistical significance was considered to be $P < 0.05$. Accordingly, the restricted cubic spline (RCS) curves were plotted to visualize the associations of UI with BMI by sex in HRS, ELSA, and SHARE, as well as the associations of UI with WC by sex in HRS rather than in ELSA and SHARE due to only one wave for WC statistics in ELSA and data unavailability of WC in SHARE.

Additionally, subgroup analyses were performed for each age group (50–59/60–69/70–79/80 over) within each gender (female/male). The RCS curves were shown for relationships between UI and BMI for each age group in HRS and SHARE, as well as the associations between UI and WC for each age group in HRS. The subgroup analyses in ELSA, however, were not conducted because there was no significantly discernible gender difference in the associations. All statistical analyses were performed by Stata 17.0. The key codes for this analysis have been made available online as Supplementary Data 3.

**Reporting summary**. Further information on research design is available in the Nature Portfolio Reporting Summary linked to this article.

## Results

The descriptive characteristics of observations by sex in the HRS, ELSA, and SHARE are presented in Table 1. The 666,763 participant observations (HRS 207,805; ELSA 98,158; SHARE 360,800) were collected for the baseline survey. The median ages included in this analysis for HRS, ELSA, and SHARE studies were 66, 67, and 64 for females, and 65, 67, and 64 for males, respectively. The white was 72% in HRS and 95.4% in ELSA for females, as well as 73.6% in HRS and 95.4% in ELSA for males. More importantly, the females and males with UI were responsible for 15.6% and 6.6% in the HRS study, 10.6% and 4.4% in the ELSA study, and 2.8% and 1.4% in the SHARE study, respectively. The average BMI in the surveys of HRS, ELSA, and SHARE was 28.6 (±6.5), 28.20 (± 5.6), and 26.5 (±4.8) for females, and 28.5 (±5.3), 28.2 (±4.4), and 27.0 (±4.0) for males, respectively. Meanwhile, the average WC for females and males were 99.3 (±14.5) and 105.2 (±12.4) in HRS and 92.0 (±13.6) and 102.2 (±11.9) in ELSA.

The BMI, WC, and covariates of observations by sex in analyses were displayed in Supplementary Data 1. Across three cohort studies, there are 29,187 female observations with UI and 10,241 male observations with UI being incorporated. Notably, males and females with and without UI exhibited significant

**Table 1 Baseline characteristics of study subjects broken down by sex in each of the three datasets (HRS, ELSA, and SHARE)**

| Characteristics | HRS (N = 207,805) | | ELSA (N = 98,158) | | SHARE (N = 360,800) | |
|---|---|---|---|---|---|---|
| | Female (N = 115,849) | Male (N = 91,956) | Female (N = 53,325) | Male (N = 44,833) | Female (N = 198,788) | Male (N = 162,012) |
| Age (years) | 66 (58,76) | 65 (58,75) | 67 (60, 75) | 67 (61, 74) | 64 (57, 73) | 64 (57, 72) |
| White (%) | 83,372 (72.0%) | 67,635 (73.6%) | 50,857 (95.4%) | 42,776 (95.4%) | – | – |
| **Educational attainments (%)** | | | | | | |
| Less than secondary | 17,590 (15.2%) | 15,178 (16.5%) | 19,469 (36.5%) | 13,181 (29.4%) | 91,628 (46.1%) | 62,338 (38.5%) |
| Upper secondary and vocational training | 36,638 (31.6%) | 44,463 (48.4%) | 21,542 (40.4%) | 20,322 (45.3%) | 69,998 (35.2%) | 62,676 (38.7%) |
| Tertiary | 16,088 (13.9%) | 14,082 (15.3%) | 6613 (12.4%) | 8015 (17.9%) | 37,162 (18.7%) | 36,998 (22.8%) |
| **Residence area (%)** | | | | | | |
| Rural | 28,101 (24.3%) | 23,160 (25.2%) | – | – | 45,775 (23.0%) | 39,663 (24.5%) |
| Urban | 78,431 (67.7%) | 61,214 (66.6%) | – | – | 102,982 (51.8%) | 83,292 (51.4%) |
| Married and partnered (%) | 61,393 (53.0%) | 27,527 (29.9%) | 20,607 (38.6%) | 11,283 (25.2%) | 58,966 (29.7%) | 27,708 (17.1%) |
| Number of Children (n) | 2 (1,4) | 2 (1,4) | 2 (1,3) | 2 (1,3) | 2 (1,3) | 2 (1,3) |
| Urinary incontinence (%) | 18,070 (15.6%) | 6065 (6.6%) | 5629 (10.6%) | 1973 (4.4%) | 5488 (2.8%) | 2203 (1.4%) |
| Current smoking (%) | 7148 (6.2%) | 6175 (6.7%) | 2817 (5.3%) | 2387 (5.3%) | 10,286 (5.2%) | 12,422 (7.7%) |
| Ever smoked (%) | 27,041 (23.3%) | 25,978 (28.3%) | 14,419 (27.0%) | 14,141 (31.5%) | 22,454 (11.3%) | 34,081 (21.0%) |
| Alcohol consumption (%) | 27,408 (23.7%) | 25,528 (27.8%) | 18,412 (34.5%) | 15,956 (35.6%) | 37,284 (18.8%) | 36,532 (22.5%) |
| Physically active (%) | 50,811 (43.9%) | 38,153 (41.5%) | 24,689 (46.3%) | 20,305 (45.3%) | 65,096 (32.7%) | 53,100 (32.8%) |
| **History of diseases** | | | | | | |
| Hypertension (%) | 33,872 (29.2%) | 24,563 (26.7%) | 10,802 (20.3%) | 9490 (21.2%) | 27,516 (13.8%) | 20,544 (12.7%) |
| Diabetes (%) | 13,547 (11.7%) | 10,716 (11.7%) | 2607 (4.9%) | 2988 (6.7%) | 7873 (4.0%) | 7163 (4.4%) |
| Cancer (%) | 8127 (7.0%) | 6253 (6.8%) | 3203 (6.0%) | 2272 (5.1%) | 4662 (2.3%) | 3300 (2.0%) |
| Stroke (%) | 4981 (4.3%) | 3850 (4.2%) | 1162 (2.2%) | 1159 (2.6%) | 3017 (1.5%) | 3032 (1.9%) |
| Cognitive impairment (%) | 5110 (4.4%) | 4677 (5.1%) | 1736 (3.3%) | 1264 (2.8%) | 7738 (3.9%) | 5532 (3.4%) |
| BMI (kg/m$^2$) | 28.6 ± 6.5 | 28.5 ± 5.3 | 28.2 ± 5.6 | 28.2 ± 4.4 | 26.5 ± 4.8 | 27.0 ± 4.0 |
| Waist circumferences (cm) | 99.3 ± 14.5 | 105.2 ± 12.4 | 92.0 ± 13.6 | 102.2 ± 11.9 | – | – |

Data are presented as mean ± SD or median (IQR), n (%).
*HRS* the Health and Retirement Study, *ELSA* the English Longitudinal Study of Ageing, *SHARE* Survey of Health, Ageing and Retirement in Europe.
*BMI* body mass index.

differences in baseline characteristics. Specifically, females with UI showed significantly higher ($Ps < 0.05$) median age, the proportion of white, married and partnered, and physically inactive, the prevalence of chronic diseases and cognitive impairments (except HRS study), and higher BMI and WC, compared to females without UI. Comparably, most baseline characteristics for males with and without UI indicated similar trends with females but with less statistical significance in race, residence area, marital status, the number of children, variables for lifestyle, and obesity indices.

The association between BMI, WC, and the prevalence of UI was outlined in Supplementary Data 2. The fully adjusted model for females shows a linear trend of monotonically increasing prevalence of UI as BMI and WC rise. Specifically, the first quantile had the lowest prevalence of UI, and the sixth quantile had the significantly highest prevalence of UI ($Ps < 0.001$) compared to participants in the second quantile of BMI. Similarly, compared to participants in the second quantile of WC, those in the first quantile had the lowest prevalence of UI, whereas the significant highest prevalence of UI ($Ps < 0.001$) was observed in the sixth quantile.

However, a U-shaped trajectory in the UI prevalence can be found with rising BMI and WC in the fully adjusted model for males, even though the linear associations between obesity indices and UI were less pronounced than in females. In comparison to those in the second quantile of BMI, the lowest prevalence of UI was found in the fourth quantile for HRS, the third quantile for ELSA and SHARE study, and those in the sixth quantile of BMI had the highest prevalence of UI. In terms of WC, we discovered that those in the fourth quantile of WC for both HRS and ELSA experienced the lowest prevalence of UI. However, those statistics were not statistically significant. Additionally, although the statistics in the ELSA study did not reach statistical significance, the statistic for the HRS in the sixth quantile of WC had the significantly highest UI prevalence ($Ps < 0.01$) compared to those in the second quantile.

Additionally, we draw the restricted cubic splines (RCS) curve to visualize the association between UI and BMI by sex in HRS, ELSA, and SHARE, as shown in Fig. 1. Besides, the association between UI and WC by sex in the HRS study was also plotted in Fig. 2. The source datasets underlying the figures are available from the GATEWAY TO GLOBAL AGING DATA (https://g2aging.org/) upon registration and request. The shapes of RCS curves showed heterogeneity across genders in HRS and SHARE studies. Specifically, among three longitudinal cohort studies, the RCS curves for females concerning the association of UI with both BMI and WC were almost monotonically increasing, and the slopes of these curves were relatively larger at higher BMI and higher WC. However, for males in HRS and SHARE, it was obvious that a "U-" shaped curve was traced with increasing OR of the prevalence of UI on the y-axis and increasing BMI or increasing WC on the x-axis, even though this pattern was not apparent for the figure in ELSA study.

To assess the gender and age heterogeneity of UI on BMI and WC, RCS curves were also drawn in Supplementary Fig. 1, Supplementary Fig. 2, and Supplementary Fig. 3 to visualize the UI-BMI association and UI-WC association in different age groups within each gender in HRS and SHARE. As shown in Supplementary Table 1, subgroup analysis was conducted to assess the gender and race heterogeneity of UI on BMI and WC in HRS. As can be seen, we found that both the UI–BMI association and the UI–WC association were more pronounced in the 60–69 age group for males in HRS and SHARE.

## Discussion
By utilizing three longitudinal cohorts representative of the US, UK, and European samples, this study offered unique insight and

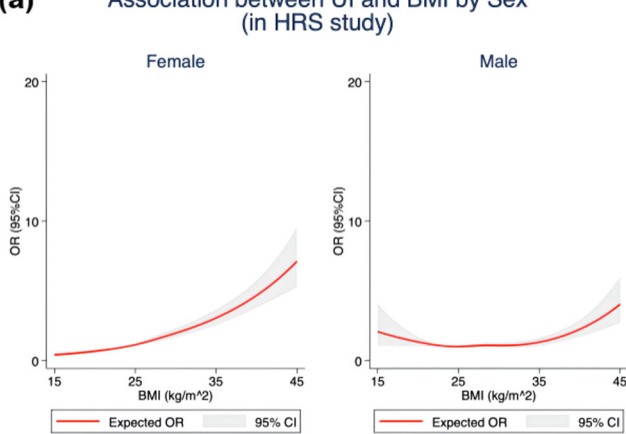

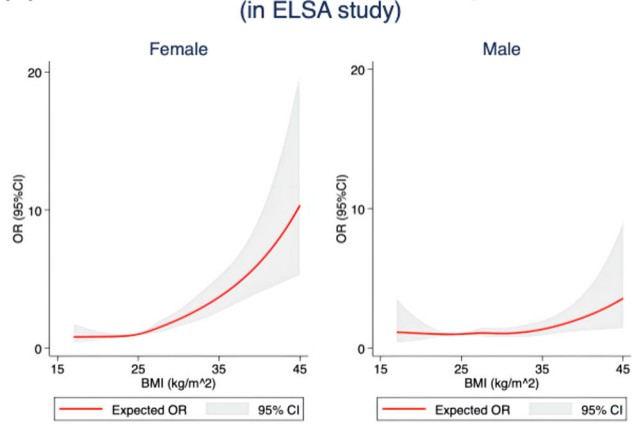

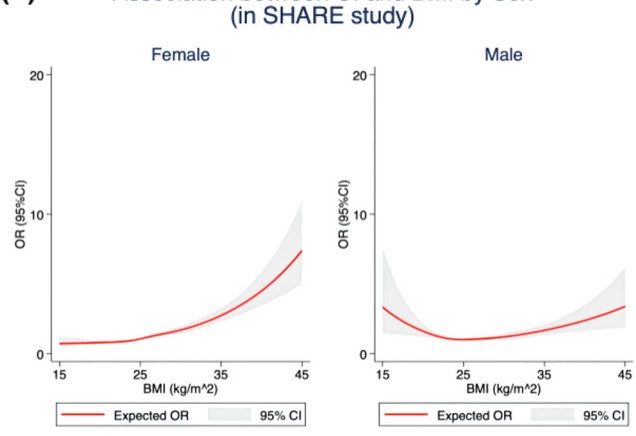

**Fig. 1 Association between UI and BMI by sex in HRS, ELSA, and SHARE. a** Associations between UI and BMI by sex in HRS study. **b** Associations between UI and BMI by sex in ELSA study. **c** Associations between UI and BMI by sex in SHARE study. Red solid lines expected odds ratio; Gray shades, 95% confidence interval. HRS the Health and Retirement Study, ELSA the English Longitudinal Study of Ageing, SHARE Survey of Health, Ageing and Retirement in Europe, UI urinary incontinence, BMI body mass index, OR odds ratio, 95% CI 95% confidence interval.

extensive evidence on the relationship between obesity and UI in older men and women. Specifically, the higher prevalence of UI in older women is linearly correlated with higher BMI and WC, while there was a U-shaped association between BMI and UI in older men. Our study indicated a revisiting of the

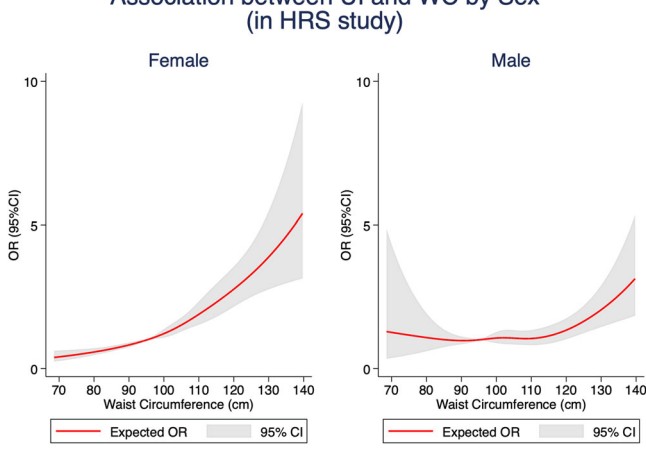

**Fig. 2 Association between UI and WC by sex in HRS, 2010–2018.** Red solid lines expected odds ratio; Gray shades, 95% confidence interval. HRS the Health and Retirement Study, UI urinary incontinence, WC waist circumferences, OR odds ratio, 95% CI 95% confidence interval.

association between obesity and UI among older men compared to women.

Prior epidemiological studies reported that obesity is a modifiable risk factor for UI development in older women or men[5–8,10,11,14,15,17,37], and some clinical trials provided evidence that weight reduction can reduce the incidence of UI[9,12,13,16]. However, only seven studies—three longitudinal and four cross-sectional—focused on men and women separately regarding the relationship between UI and obesity indices[18,20–25]. In particular, Subak et al.'s cohort study based on the longitudinal assessment of bariatric surgery 2 revealed that participants with obesity who lose weight via surgical interventions experience reduced UI, which was generally in line with the results of our study. However, the target research subjects are diverse between these two studies. Specifically, Subak et al.'s study focused on people with severe obesity with BMI ($kg/m^2$) varied from 41.6 to 52.8, whereas our study subjects concentrated on the general elderly population[23].

In addition, the cohort study by Tsui et al. from a British birth cohort study demonstrated that increased BMI at age 60–64 years was an independent risk factor for urgent urinary incontinence (UUI) in men and women[25,38]. Another article by Tennstedt et al. examining the population-based adults aged 30–79 years based on the Boston Area Community Health (BACH) Survey investigated that WC was a risk factor for leakage in women rather than men, with the odds of weekly leakage increasing by 15 percent with each 10-cm increase[24,39]. In general, our findings concurred with these investigations. However, the participants in these two research specifically targeted the elderly in Britain and Boston, respectively, rather than the elderly worldwide. Second, only UUI, rather than general UI, was used as the outcome assessment in the Tsui et al. study. Furthermore, we could only be enlightened if obesity was related to UUI or UI rather than the trajectory of UI as WC grew and the tendency of UUI with growing BMI.

Moreover, even though there were four cross-sectional studies also considering both genders in regard to the UI-obesity associations[18,20–22,40], one of them from the French 3 C study that concentrated on the elderly subjects aged 65–101 suggested that the relationship tended to be linear for UI and obesity in females, whereas the pattern that a higher risk of UI for both subjects who are underweight and with obesity was found in males[22,41]. This study was in line with our findings but

concentrated primarily on French and did not take into account the obesity indices for central adiposity.

In terms of mechanisms, several biological studies have pointed out that obesity predisposes individuals to a wide range of chronic diseases, such as insulin resistance, diabetes, and lower urinary tract symptoms, which are closely related to low-grade systemic inflammation and oxidative stress. At the same time, intra-abdominal pressure increases with obesity, causing increased bladder pressure and urethral mobility, as well as chronic strain, stretching, and weakening of the muscles, nerves, and other structures of the pelvic floor, which thus leads to negatively affecting pelvic organ function. This series of events potentially lead to SUI, as well as aggravate symptoms of detrusor instability, overactive bladder, and urgency urinary incontinence. These mechanical and biochemical stresses predispose geriatric patients with obesity to develop UI[8,42]. Comparably, being underweight also resulted in a higher prevalence of UI for males, according to our study. The "U-shaped trajectory" in males might be explained by confounding by an unmeasured covariate, such as frailty. Specifically, being underweight may cause frailty that could confer a higher risk of falls, disability, and hospitalization[43]. Nonetheless, the association between being underweight and UI is still debatable and appears to be weaker than the relationship with obesity. The discovery of permanent rather than reversible alterations to the prostate caused by weight loss once men approach old age[11], as well as postmenopausal hormonal changes in women, maybe viable explanations in light of the mechanism underlying the gender difference. Additionally, the type of UI is different for males and females, with males having a higher proportion of urgency and overflow UI likely due to benign prostatic hyperplasia (BPH). The type of incontinence, however, was not taken into account in this study's outcome measures, which was likely the cause of the less substantial linear association in men compared to women.

This study delivers meaningful policy implications. On the one hand, the relationship between UI and obesity indices in males has been insufficiently underappreciated and incompletely understood. For instance, earlier research has focused less on the patterns between UI and obesity in males than in females. Male participants were not even asked in the Korean Longitudinal Study on Aging (KLoSA) questionnaire if they had ever experienced UI within a certain time period. Moreover, we have an incomplete understanding of male UI and obesity and meaningful evidence that we should not generalize our understanding of UI and obesity in females and extrapolate that to males. Overall, although males are less likely to report UI than females, it is nevertheless essential to revisit the UI-obesity association in males to better corroborate our findings and thus further improve our present UI prevention and treatment practices. On the other hand, the majority of developed countries examine UI and fecal incontinence separately due to crucial diversity in pathogenic mechanisms, risk factors, and treatment approaches, which HRS, ELSA, KLoSA, and SHARE could support. Contrarily, some nationwide cohort studies of aging, including the China Health and Retirement Longitudinal Study (CHARLS)[44], did not differentiate these two disorders clearly and instead created a single question to target them explicitly. Therefore, it was imperative to distinguish UI and fecal incontinence when developing survey questionnaires to advance pertinent intervention research.

There are several strengths. Firstly, our study's excellent generalizability can be attributed to the utilization of three sizable, diverse, well-characterized, cross-cultural longitudinal studies that included older adults from 22 developed and developing countries on two continents. Second, the validity of our study was strengthened by the use of nationally representative samples for participant recruitment, standardization of the three surveys for

database comparisons, and a more extended follow-up period of almost ten years. Third, random-effect logistic models regulate individual-level traits that could alter over longitudinal data waves, thus reducing the likelihood of misestimation. Additionally, most previous studies focused on the patients, whereas we used a large sample of the community-based population with a higher level of representativeness. Hence, the strengths of our study are that the results partly corroborate previous studies and take extrapolation into account. Finally, this study provides a basis for demonstrating the obesity paradox in the relationship between obesity and urinary incontinence in men, showing that older men with both lower and higher BMI and WC tend to report higher UI prevalence compared to those with BMI or WC in the normal range, although this relationship is not particularly significant.

However, our study has limitations. Firstly, even though we considered numerous potential confounders that had a vital impact on our findings, there may still be confounding variables that are unaccounted for due to limited information, such as ethnicity in SHARE and residential areas in ELSA. A previous study provided insights that indicators of central obesity, including WC and waist-to-hip ratio, appear to be more sensitive than BMI (a proxy for body fat) in explaining the association between obesity and urine leakage[21,24]. However, our study did not fully consider the indicators of central obesity due to a lack of comparable data. Besides, the type of incontinence was not specified in the longitudinal cohort, which poses an unavoidable obstacle to further research on its mechanisms, and we believe follow-up studies are needed to analyze the specific association and mechanisms between obesity indices and UI by different UI types in depth, as well as more detailed subgroup analyses including racial, residence area and etc.

In summary, the associations between obesity indices and UI are different in older women compared to older men. Therefore, weight loss as a treatment for UI can only be applied to older women but not necessarily to men. With this in mind, developing interventions to address UI among older adults should take sex into account.

### Data availability

The original survey datasets used and analyzed are publicly available which are also provided with this paper. These datasets that support the findings of this study are available from the GATEWAY TO GLOBAL AGING DATA (https://g2aging.org/) upon registration.

### Code availability

The key codes for this analysis have been made available online as Supplementary Data 3.

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

## Acknowledgements

We thank the GATEWAY TO GLOBAL AGING DATA for providing the harmonized data. We thank Dr. Rui Yang and Mingzhi Yu from Peking University for giving us technical support in data analyses and interpretation. We also thank the support from the Healthy Aging Group of the China Cohort Consortium (see http://chinacohort.bjmu.edu.cn/).

## Author contributions

Y.Y. designed the study. Data collection and analysis was led by X.C. The paper was written by X.C. and verified by S.J. All authors have edited and reviewed the paper. All authors had full access to the data and accepted the responsibility to submit the paper.

## Funding

This research was supported by the National Natural Science Foundation of China (72374013).

## Competing interests

The authors declare no competing interests.
