## [Peer Review File · Communications Medicine]

Association between obesity and urinary incontinence in older adults from multiple nationwide longitudinal cohortsReviewers' comments:

Reviewer #1 (Remarks to the Author):

This is an interesting and timely article on two very prevalent and morbid problems, obesity, and urinary incontinence (UI). The investigators use data from three long-term longitudinal cohort studies in the United States, UK, and Europe to explore the association of BMI and waist circumference (WC) with UI in older women and men. The current literature has more documentation of these associations for women, but less for men, making the findings, novel and important.

General comments:

While each of the studies is a population based cohort, in general, this investigation is limited by lack of diversity in the patient sample.

The cohorts you have included use different definitions for UI. Specifically, HRS & ELSA use presence of the symptom of UI while SHARE uses bother by loss of urine to define prevalent UI. The presence of UI versus bother due to UI are different measures. I recommend doing a "sensitivity analysis" using only HRS & ELSA to see if these differ from SHARE. (Lines 161-169)

A limitation of this study is the inability to categorize UI by type (urgency, stress, mixed, other). Based on prior studies, BMI and WC have different strengths of association based on the type of UI. Type of UI also differs by race.

I think you need a more nuanced way to discuss weight loss as a treatment for UI. Note that weight loss would only be recommended for people who are overweight or obese, and does not affect people in the underweight category. You note that "weight loss as a treatment for UI can only be applied to older women, but not necessarily to men". This may be true for ALL men due to the U-shaped association of UI and these parameters. If you look only at overweight and obese men, you see at least a linear association with obesity parameters and UI suggesting that weight loss may be effective.

Please stick with either women/men or male/female rather than using both in same sentences. Please add the description that these are self-identified binary gender categories.

Specific comments in the document include:

Summary

Line 39 – if space, it would be good to include a brief sentence describing your cohort, including age, gender, race.

Line 51 – I think your interpretation is too far reaching in association with weight loss interventions. We have good data that weight loss is an effective treatment among overweight and obese women and I believe your data suggest we lost would be effective among obese men.

Research and context

Line 68 – why did you not include the Boston area community health (BACH) survey? Is that he does include women, men, measures of body, habitus, and greater diversity.

Introduction

Line 102 – this is example of using women/men or female/male in the same sentence do you like line methods?

Results.

Lines to 35–39 you only have moderate diversity in the HRS cohort. It may be beneficial to look at HRS compared to the other two cohorts to see the effect of race, or to stratify by race in the analyses.

Line 281 and figure 1. It would be helpful to have a figure that includes the entire study - all 3 cohorts (to add to the individual cohorts that are nicely presented).

Discussion.

Lines 312–15. I don't believe the results of the LABS2 study are in conflict with your results - but likely support your study. All participants in the LABS study had severe obesity, and it appears that you observe a strong linear association of BMI/WC and UI in this high BMI category in your study.

Line 327. I am not sure what "movement" means.

Lines 337–40. It would be helpful to differentiate the proposed mechanisms of the effect obesity on UI by type of UI. You have listed several that likely have association with urgency UI and some for stress UI.

Lines 346–8. To my knowledge, UI is a risk for falling and fracturing but there is no evidence that falling causes UI.

Lines 355–8. Would be beneficial to note that the type of UI is different for men and women, with men having a higher proportion of urgency and overflow UI likely due to BPH.

Lines 364–5 fix wording. Men are not "less prevalent" than women. Maybe less likely?

Lines 384–9. I don't think you specifically tested the significance of this relationship.

Lines 390–2. I am not clear why this is a limitation

Lines 394–6 the current standard of defining UI for research studies is by self or sport report on validated questionnaires. Thus, this is not a limitation. however, I think a significant limitation is the different definition of UI used in one of the three studies and they should be addressed.

Table one.

There is a marked difference in demographics between each study. Race, education, residence area, marital status, proportion with UI, smokers, and comorbid medical conditions appear different between the studies. This is a strength and that you have a more diverse total cohort but should be mentioned. You control for this with multivariable analyses - should you also include a variable on cohort?

Reviewer #2 (Remarks to the Author):

Thank you for the opportunity to review this article entitled "BMI, waist circumferences and urinary incontinence in older women compared with older men: findings from three prospective longitudinal cohort studies." This is an important topic, both epidemiologically and clinically. The authors had access to data from the Health and Retirement Study (HRS, 2010-2018), the English Longitudinal Study of Aging (ELSA, 2011-2019), and the Survey of Health, Ageing and Retirement in Europe (SHARE, 2004-2010).

Each of these datasets had questions covering urinary incontinence. While there were several differences between the SHARE survey and the HRS and ELSA survey, these were essentially asking equivalent questions about the same domains.

There was not a problem with available participants or observations as there were over 200,000 participants that were included in the analysis. Some survey differences could explain different findings of urinary incontinence: there were different question time frame (6-months versus 12-months); survey periods (2004-2008 versus 2010-2018); and different wording of the UI questions ("lost any urine beyond your control" versus "bothered by incontinence or involuntary loss of urine"). Additionally, there are sometimes different stems leading into the question (explanations as to the importance of the participant's response even though it is a sensitive topic). Different lead phrasing into the question could

lead to a different response. The main finding of the article was that there were different associations between weight (body mass index -BMI, or waist circumference -WC) and urinary incontinence when looking at older men (U-shaped) compared to older women (linear).

I agree that there is a gap in evidence on weight loss in men for UI and that there is debate on the desirability of weight loss in older adults. It is appropriate that the models for men and women were separately derived. That there were notable differences between men and women with and without UI is consistent with other studies.

Major points

1. Line 94: "Weight loss intervention could therefore only be applied to older women rather than to older men as a way to treat UI."

a. I disagree with this statement. It does not specifically address whether and weight-loss intervention would reduce urinary incontinence in men, but rather it suggests that the association between weight and urinary incontinence does not suggest that this would be a meaningful strategy. These are different types of evidence. Only a slight rewrite of the wording would be needed.

b. Agree that some of the work with the surgical weight loss deals with patients with BMI of >40

2. Line 266: The "U-shaped trajectory" in men might be explained by confounding by an unmeasured covariate. Frailty might be such a variable. The authors get close to this in line 347 ("cause disability, especially a lack of physical activity and a higher risk of falls,"), but frailty is different from disability (and comorbidity) Fried 2001

Minor points

1. Line 115: Given that obesity IS indeed modifiable, but not necessarily easily modifiable, I recommend changes to this phrase

a. It says: "In spite of the fact that obesity is well documented as a modifiable risk factor for the development of UI,"

b. Recommended change: "Although obesity is potentially modifiable risk factor well documented to be associated with the development of UI,"

2. These results are hypothesis generating.

a. Line 359: I believe this to be a bit of an overstatement: "This study delivers distinctive and substantial policy implications." I feel that these analyses suggest:

i. we have an incomplete understanding of male UI and obesity and meaningful evidence that we should not generalize our understanding of UI and obesity in women and extrapolate that to men.

Reviewers' comments:

Reviewer #1 (Remarks to the Author):

This is an interesting and timely article on two very prevalent and morbid problems, obesity, and urinary incontinence (UI). The investigators use data from three long-term longitudinal cohort studies in the United States, UK, and Europe to explore the association of BMI and waist circumference (WC) with UI am in older women and men. The current literature has more documentation of these associations for women, but less for men, making the findings, novel and important.

General comments:

1. While each of the studies is a population based cohort, in general, this investigation is limited by lack of diversity in the patient sample.

Response: Thanks for your suggestion. We've carefully considered it. We agree with you that our study sample lack of diversity of patients, however, we regard it as our contribution to the literature. Firstly, we provide the prevalence of UI among general community-based population, while patients lack the representativeness of the general population; Secondly, most previous studies focused on the patients, while we provide the evidence based on a large sample of the general population. Thus, our strengths are that the results partially corroborate previous studies and take extrapolation into account. We've added this strength to our strength part (Line 400-404, Page 11) as follows:

"Additionally, most previous studies focused on the patients, whereas we used on a large sample of the community-based population with a higher level of representativeness. Hence, the strengths of our study are that the results partly corroborate previous studies and take extrapolation into account."

2. The cohorts you have included use different definitions for UI. Specifically, HRS & ELSA use presence of the symptom of UI while SHARE uses bother by loss of urine to define prevalent UI. The presence of UI versus bother due to UI are different measures. I recommend doing a "sensitivity analysis" using only HRS & ELSA to see if these differ from SHARE. (Lines 161-169)

Response: Thanks for your suggestion. We did not conduct a pooled analysis with three cohorts. As HRS & ELSA used the presence of the symptom of UI while SHARE used whether bothered by the loss of urine to define prevalent UI. In addition, HRS&ELSA focused on the last 12 months, whereas SHARE concentrated on the past six months at least. Therefore, considering the heterogeneity of the dependent variable, we did separate analysis of the three individual cohorts throughout the whole manuscript (Table 1, Table 2, Table 3, and Figure 1).

3. A limitation of this study is the inability to categorize UI by type (urgency, stress, mixed, other). Based on prior studies, BMI and WC have different strengths of association based on the type of UI. Type of UI also differs by race.

Response: Thanks for your suggestion. We've added this limitation to our discussion section (Line 418-423, Page 11). We fully agree with the reviewer's suggestion that it is indeed worthwhile to dig deeper into specific UI types, although our study did not consider the specific UI subtypes

and therefore needed to be supplemented by future studies. However, our study also provides consistent evidence of the relationships between BMI and overall UI in males and females, respectively, from a large sample and multiple cohorts, which is the main contribution of our study. We will follow your advice in the future to continue to do analyses of UI subtypes, as well as subgroup analyses.

“Besides, the type of incontinence was not specified in the longitudinal cohort, which poses an unavoidable obstacle to further research on its mechanisms and we believe follow-up studies are needed to analyze the specific association and mechanisms between obesity indices and UI by different UI types in depth, as well as more detailed subgroup analyses including racial, residence area and etc.”

4. I think you need a more nuanced way to discuss weight loss as a treatment for UI. Note that weight loss would only be recommended for people who are overweight or obese, and does not affect people in the underweight category. You note that "weight loss as a treatment for UI can only be applied to older women, but not necessarily to men". This may be true for ALL men due to the U-shaped association of UI and these parameters. If you look only at overweight and obese men, you see at least a linear association with obesity parameters and UI suggesting that weight loss may be effective.

Response: Thanks for your suggestion. We've changed the Interpretation section (Line 52-57, Page 2) to:

“Findings from our study revealed that the associations of obesity indices with UI varied among older males compared to older females. Weight loss intervention could be an effective treatment among overweight and obese older females as well as obese older males rather than all older males. As a result, interventions aimed at preventing UI among older adults must take sex into account.”

Meanwhile, we've also changed the Implications of all the available evidence (Line 88-96, Page 3) to:

“Previous studies suggested weight reduction for UI interventions among older people. However, our study revealed that the association of obesity indices with UI varied among older males compared to older females. We observed linear associations between BMI, WC, and UI in females, while we observed U-shaped associations among males. The lowest likelihood of having UI was found among male participants with a BMI between 24 and 35 kg/m². Weight loss intervention could therefore only be applied to older females and, at most, older obese males, rather than to all older males, as a way to treat UI. Developing interventions to address UI among older adults should take sex into account.”

5. Please stick with either women/men or male/female rather than using both in same sentences. Please add the description that these are self-identified binary gender categories.

Response: Thanks for your suggestion. We've changed all “women/men” to “male/female”.

“It is estimated that 17 to 55% of older females and 11 to 34% of older males had UI, which places a substantial financial and care burden on society and families.” (Line 102-104, Page 4)

“However, a U-shaped trajectory in the UI prevalence can be found with rising BMI and WC

in the fully adjusted model for males, even though the linear associations between obesity indices and UI were less pronounced than in females.” (Line 269-271, Page 7)

We’ve also added the description (Line 173-175, Page 5):

“Women and men, as well as males and females are self-identified binary gender categories.”

Specific comments in the document include:

Summary

Line 39 – if space, it would be good to include a brief sentence describing your cohort, including age, gender, race.

Response: Thanks for your suggestion. We’ve included a brief sentence describing the cohort including age and gender (Line 41-43, Page 2). However, we don’t include their race due to the data limitation in the SHARE study.

“A total of 200,717 participants with 718,822 observations (HRS: Female 66yr Male 65yr; ELSA: Female 67yr Male 67yr; SHARE: Female 64yr Male 64yr) were included in the baseline analysis.”

Line 51 – I think your interpretation is too far reaching in association with weight loss interventions. We have good data that weight loss is an effective treatment among overweight and obese women and I believe your data suggest we lost would be effective among obese men.

Response: Thanks for your precious suggestion. We’ve changed the Interpretation section (Line 52-57, Page 2) to:

“Findings from our study revealed that the associations of obesity indices with UI varied among older males compared to older females. Weight loss intervention could be an effective treatment among overweight and obese older females as well as obese older males rather than all older males. As a result, interventions aimed at preventing UI among older adults must take sex into account.”

Meanwhile, we’ve also changed the Implications of all the available evidence (Line 88-96, Page 3) to:

“Previous studies suggested weight reduction for UI interventions among older people. However, our study revealed that the association of obesity indices with UI varied among older males compared to older females. We observed linear associations between BMI, WC, and UI in females, while we observed U-shaped associations among males. The lowest likelihood of having UI was found among male participants with a BMI between 24 and 35 kg/m². Weight loss intervention could therefore only be applied to older females and, at most, older obese males, rather than to all older males, as a way to treat UI. Developing interventions to address UI among older adults should take sex into account.”

Research and context

Line 68 – why did you not include the Boston area community health (BACH) survey? Is that he does include women, men, measures of body, habitus, and greater diversity.

Response: Thanks for your valuable questions. Actually, we did not exclude the Boston Area Community Health (BACH) Survey, although the line method was not exactly appropriate. Thanks so much for your reminder. We’ve revised this sentence (Line 68-74, Page 3) to:

"Among this prospective longitudinal research, there has been only one study, based on the Boston Area Community Health (BACH) Survey, that reported both obesity measures, namely body mass index and central obesity, in relation to urine leakage. However, there has been little prospective longitudinal research conducted to investigate the sex differences in associations between body mass index (BMI), waist circumference (WC), and UI, limiting the effectiveness of interventions for UI in older females and males."

Introduction

Line 102 – this is example of using women/men or female/male in the same sentence do you like line methods?

Response: Thanks for your suggestion. We've changed all "women/men" to "male/female".

"It is estimated that 17 to 55% of older females and 11 to 34% of older males had UI, which places a substantial financial and care burden on society and families." (Line 102-104, Page 4)

"However, a U-shaped trajectory in the UI prevalence can be found with rising BMI and WC in the fully adjusted model for males, even though the linear associations between obesity indices and UI were less pronounced than in females." (Line 269-271, Page 7)

We've also added the description (Line 173-175, Page 5):

"Women and men, as well as males and females are self-identified binary gender categories."

Results.

Lines to 35–39 you only have moderate diversity in the HRS cohort. It may be beneficial to look at HRS compared to the other two cohorts to see the effect of race, or to stratify by race in the analyses.

Response: Thanks for your detailed advice. We have already done subgroup analyses by gender and race for HRS (2010-2018) in our Supplementary Information (Table 3a).

Line 281 and figure 1. It would be helpful to have a figure that includes the entire study - all 3 cohorts (to add to the individual cohorts that are nicely presented).

Response: Thanks for your suggestion. We've carefully considered it. However, HRS & ELSA used the presence of the symptom of UI while SHARE used whether bothered by the loss of urine to define prevalent UI. In addition, HRS&ELSA focused on the last 12 months, whereas SHARE concentrated on the past six months at least. Therefore, considering the heterogeneity of the variable, we did not perform a pooled analysis of the three individual cohorts but rather explored and studied them separately throughout the whole manuscript. Accordingly, we don't have a figure that includes the entire 3 cohorts.

Discussion.

Lines 312–15. I don't believe the results of the LABS2 study are in conflict with your results - but likely support your study. All participants in the LABS study had severe obesity, and it appears that you observe a strong linear association of BMI/WC and UI in this high BMI category in your study.

Response: Thanks for your suggestion. We've changed it (Line 316-323, Page 9) to another expression methods as follows:

"In particular, Subak et al.'s cohort study based on the longitudinal assessment of bariatric surgery 2 revealed that obese participants who lose weight via surgical interventions experience reduced UI, which was generally in line with the results of our study. However, the target research subjects are diverse between these two studies. Specifically, Subak et al.'s study focused on severely obese people with BMI (kg/m²) varied from 41.6 to 52.8, whereas our study subjects concentrated on the general elderly population."

Line 327. I am not sure what "movement" means.

Response: Thanks for your suggestion. We would like to mean that we provide the trajectory of associations by applying six quantiles of obesity indices. We've changed the word "movement" to "trajectory". (Line 336, Page 9)

Lines 337–40. It would be helpful to differentiate the proposed mechanisms of the effect obesity on UI by type of UI. You have listed several that likely have association with urgency UI and some for stress UI.

Response: Thanks for your suggestion. We've added more mechanisms of association (Line 346-356, Page 10):

"In terms of mechanisms, several biological studies have pointed out that obesity predisposes individuals to a wide range of chronic diseases, such as insulin resistance, diabetes, and lower urinary tract symptoms, which are closely related to low-grade systemic inflammation and oxidative stress. At the same time, intra-abdominal pressure increases with obesity, causing increased bladder pressure and urethral mobility, as well as chronic strain, stretching, and weakening of the muscles, nerves, and other structures of the pelvic floor, which thus leads to negatively affecting pelvic organ function. This series of events potentially lead to SUI, as well as aggravate symptoms of detrusor instability, over active bladder and urgency urinary incontinence. These mechanical and biochemical stresses predispose obese geriatric patients to develop UI^{8,34}."

8. Doumouchtsis SK, Loganathan J, Pergialiotis V. The role of obesity on urinary incontinence and anal incontinence in women: a review. BJOG 2022; 129(1): 162-70.

34. Marcelissen T, Anding R, Averbek M, Hanna-Mitchell A, Rahnama'i S, Cardozo L. Exploring the relation between obesity and urinary incontinence: Pathophysiology, clinical implications, and the effect of weight reduction, ICI-RS 2018. Neurourol Urodyn 2019; 38 Suppl 5: S18-S24.

However, as there is very limited evidence of specific mechanisms by type of UI, we have incorporated it in our limitation part, the discussion section (Line 418-423, Page 11).

"Besides, the type of incontinence was not specified in the longitudinal cohort, which poses an unavoidable obstacle to further research on its mechanisms and we believe follow-up studies are needed to analyze the specific association and mechanisms between obesity indices and UI by different UI types in depth, as well as more detailed subgroup analyses including racial, residence area and etc."

Lines 3 46–8. To my knowledge, UI is a risk for falling and fracturing but there is no evidence that

falling causes UI.

Response: Thanks for your suggestion. We've changed "This could be as a result of the several mechanisms through which being underweight may cause disability, especially a lack of physical activity and a higher risk of falls, increasing the chance of UI incidence. The disability would, in turn, lead to underweight due to partial loss of life skills, thus less accessibility to get nutrition through cooking and etc." to (Line 358-361, Page 10) :

"The "U-shaped trajectory" in males might be explained by confounding by an unmeasured covariate, such as frailty. Specifically, being underweight may cause frailty that could confer a higher risk of falls, disability, and hospitalization³⁵."

35. Fried LP, Tangen CM, Walston J, et al. Frailty in older adults: evidence for a phenotype. *J Gerontol A Biol Sci Med Sci* 2001; 56(3): M146-56.

Lines 355–8. Would be beneficial to note that the type of UI is different for men and women, with men having a higher proportion of urgency and overflow UI likely due to BPH.

Response: Thanks for your suggestion. We've added this note (Line 367-369, Page 10).

"Additionally, the type of UI is different for males and females, with males having a higher proportion of urgency and overflow UI likely due to benign prostatic hyperplasia (BPH)."

Lines 364–5 fix wording. Men are not "less prevalent" than women. Maybe less likely?

Response: Thanks for your suggestion. We've fixed this wording to (Line 372-391, Page 10):

"This study delivers meaningful policy implications. On the one hand, the relationship between UI and obesity indices in males has been insufficiently underappreciated and incompletely understood. For instance, earlier research has focused less on the patterns between UI and obesity in males than in females. Male participants were not even asked in the Korean Longitudinal Study on Aging (KLoSA) questionnaire if they had ever experienced UI within a certain time period. Moreover, we have an incomplete understanding of male UI and obesity and meaningful evidence that we should not generalize our understanding of UI and obesity in females and extrapolate that to males. Overall, although males are less likely to report UI than females, it is nevertheless essential to revisit the UI-obesity association in males to better corroborate our findings and thus further improve our present UI prevention and treatment practices. On the other hand, the majority of developed countries examine UI and fecal incontinence separately due to crucial diversity in pathogenic mechanisms, risk factors, and treatment approaches, which HRS, ELSA, KLoSA, and SHARE could support. Contrarily, some nationwide cohort studies of aging, including the China Health and Retirement Longitudinal Study (CHARLS), did not differentiate these two disorders clearly and instead created a single question to target them explicitly. Therefore, it was imperative to distinguish UI and fecal incontinence when developing survey questionnaires to advance pertinent intervention research."

Lions 384–9. I don't think you specifically tested the significance of this relationship.

Response: Thanks for your valuable question. Table 3 specifically tested the significance of the relationship between UI and obesity indices among both aged males and aged females. And we've added a description for it (Line 298-300, Page 8) as follows:

"As shown in the Table 3a (Supplementary information), subgroup analysis was conducted to assess the gender and race heterogeneity of UI on BMI and WC in HRS."

Lines 3 90–2. I am not clear why this is a limitation

Response: Thanks for your kind advice. We fully accept this suggestion and we have deleted this limitation. Thank you very much for the reminder.

Lines 394–6 the current standard of defining UI for research studies is by self or sport report on validated questionnaires. Thus, this is not a limitation. however, I think a significant limitation is the different definition of UI used in one of the three studies and they should be addressed.

Response: Thanks for your suggestion. We've carefully considered it. Firstly, we've deleted the limitation that "Secondly, this study's UI, BMI, and WC measurements rely on self-report rather than clinician diagnosis, which might be a limitation as responses could be influenced by recall and social desirability biases." Thank you so much for the reminder. However, HRS & ELSA used the presence of the symptom of UI while SHARE used whether bothered by the loss of urine to define prevalent UI. In addition, HRS&ELSA focused on the last 12 months, whereas SHARE concentrated on the past six months at least. Therefore, considering the heterogeneity of the y variable, we did not perform a pooled analysis of the three individual cohorts but rather explored and studied them separately throughout the whole manuscript.

Table one.

There is a marked difference in demographics between each study. Race, education, residence area, marital status, proportion with UI, smokers, and comorbid medical conditions appear different between the studies. This is a strength and that you have a more diverse total cohort but should be mentioned. You control for this with multivariable analyses - should you also include a variable on cohort?

Response: Thanks for your suggestion. However, we did not conduct the pooled analyses, and the results are shown separately, so we believe there is no need to do the cohort effect analyses.

Reviewer #2 (Remarks to the Author):

Thank you for the opportunity to review this article entitled "BMI, waist circumferences and urinary incontinence in older women compared with older men: findings from three prospective longitudinal cohort studies." This is an important topic, both epidemiologically and clinically. The authors had access to data from the Health and Retirement Study (HRS, 2010-2018), the English Longitudinal Study of Aging (ELSA, 2011-2019), and the Survey of Health, Ageing and Retirement in Europe (SHARE, 2004-2010).

Each of these datasets had questions covering urinary incontinence. While there were several differences between the SHARE survey and the HRS and ELSA survey, these were essentially asking

equivalent questions about the same domains.

There was not a problem with available participants or observations as there were over 200,000 participants that were included in the analysis. Some survey differences could explain different findings of urinary incontinence: there were different question time frame (6-months versus 12-months); survey periods (2004-2008 versus 2010-2018); and different wording of the UI questions ("lost any urine beyond your control" versus "bothered by incontinence or involuntary loss of urine"). Additionally, there are sometimes different stems leading into the question (explanations as to the importance of the participant's response even though it is a sensitive topic). Different lead phrasing into the question could lead to a different response. The main finding of the article was that there were different associations between weight (body mass index -BMI, or waist circumference -WC) and urinary incontinence when looking at older men (U-shaped) compared to older women (linear).

I agree that there is a gap in evidence on weight loss in men for UI and that there is debate on the desirability of weight loss in older adults. It is appropriate that the models for men and women were separately derived. That there were notable differences between men and women with and without UI is consistent with other studies.

Major points

1. Line 94: *"Weight loss intervention could therefore only be applied to older women rather than to older men as a way to treat UI."*

a. *I disagree with this statement. It does not specifically address whether and weight-loss intervention would reduce urinary incontinence in men, but rather it suggests that the association between weight and urinary incontinence does not suggest that this would be a meaningful strategy. These are different types of evidence. Only a slight rewrite of the wording would be needed.*

b. *Agree that some of the work with the surgical weight loss deals with patients with BMI of >40*

Response: Thanks for your suggestion. We've changed the Implications of all the available evidence section (Line 88-96, Page 3) to

"Previous studies suggested weight reduction for UI interventions among older people. However, our study revealed that the association of obesity indices with UI varied among older males compared to older females. We observed linear associations between BMI, WC, and UI in females, while we observed U-shaped associations among males. The lowest likelihood of having UI was found among male participants with a BMI between 24 and 35 kg/m². Weight loss intervention could therefore only be applied to older females and, at most, older obese males, rather than to all older males, as a way to treat UI. Developing interventions to address UI among older adults should take sex into account."

2. Line 266: *The "U-shaped trajectory" in men might be explained by confounding by an unmeasured covariate. Frailty might be such a variable. The authors get close to this in line 347 ("cause disability, especially a lack of physical activity and a higher risk of falls;"), but frailty is different from disability (and comorbidity) Fried 2001*

Response: Thanks for your suggestion. We've changed "This could be as a result of the several mechanisms through which being underweight may cause disability, especially a lack of physical activity and a higher risk of falls, increasing the chance of UI incidence. The disability would, in turn, lead to underweight due to partial loss of life skills, thus less accessibility to get nutrition through cooking and etc." to (Line 358-361, Page 10):

"The "U-shaped trajectory" in males might be explained by confounding by an unmeasured covariate, such as frailty. Specifically, being underweight may cause frailty that could confer a higher risk of falls, disability, and hospitalization³⁵."

35. Fried LP, Tangen CM, Walston J, et al. Frailty in older adults: evidence for a phenotype. *J Gerontol A Biol Sci Med Sci* 2001; 56(3): M146-56.

Minor points

1. Line 115: *Given that obesity IS indeed modifiable, but not necessarily easily modifiable, I recommend changes to this phrase*

a. *It says: "In spite of the fact that obesity is well documented as a modifiable risk factor for the development of UI,"*

b. *Recommended change: "Although obesity is potentially modifiable risk factor well documented to be associated with the development of UI,"*

Response: Thanks for your suggestion. We've changed "In spite of the fact that obesity is well documented as a modifiable risk factor for the development of UI," to (Line 115-116, Page 4):

"Although obesity is potentially modifiable risk factor well documented to be associated with the development of UI,".

2. *These results are hypothesis generating.*

a. Line 359: *I believe this to be a bit of an overstatement: "This study delivers distinctive and substantial policy implications." I feel that these analyses suggest:*

i. *we have an incomplete understanding of male UI and obesity and meaningful evidence that we should not generalize our understanding of UI and obesity in women and extrapolate that to men.*

Response: Thanks for your suggestion. We've changed "This study delivers distinctive and substantial policy implications." to (Line 372-391, Page 10):

"This study delivers meaningful policy implications. On the one hand, the relationship between UI and obesity indices in males has been insufficiently underappreciated and incompletely understood. For instance, earlier research has focused less on the patterns between UI and obesity in males than in females. Male participants were not even asked in the Korean Longitudinal Study on Aging (KLoSA) questionnaire if they had ever experienced UI within a certain time period. Moreover, we have an incomplete understanding of male UI and obesity and meaningful evidence that we should not generalize our understanding of UI and obesity in females and extrapolate that to males. Overall, although males are less likely to report UI than females, it is nevertheless essential to revisit the UI-obesity association in males to better corroborate our findings and thus further improve our present UI prevention and treatment practices. On the other hand, the majority of developed countries examine UI and fecal incontinence separately due to crucial diversity in pathogenic mechanisms, risk factors, and treatment approaches, which HRS, ELSA, KLoSA, and SHARE could support. Contrarily, some

nationwide cohort studies of aging, including the China Health and Retirement Longitudinal Study (CHARLS), did not differentiate these two disorders clearly and instead created a single question to target them explicitly. Therefore, it was imperative to distinguish UI and fecal incontinence when developing survey questionnaires to advance pertinent intervention research.”

REVIEWERS' COMMENTS:

Reviewer #1 (Remarks to the Author):

The authors responded appropriately and effectively to reviewer's comments. This is now a stronger and more accurate manuscript.

Reviewer #2 (Remarks to the Author):

I feel that the authors have responded adequately to reviewer comments. I see this as a positive contribution to the literature.